# Beyond the Benchmark: Detecting Diverse Anomalies in Videos

## Abstract

Video Anomaly Detection (VAD) plays a crucial role in modern surveillance systems, aiming to identify various anomalies in real-world situations. However, current benchmark datasets predominantly emphasize simple, single-frame anomalies such as novel object detection. This narrow focus restricts the advancement of VAD models. In this research, we advocate for an expansion of VAD investigations to encompass intricate anomalies that extend beyond conventional benchmark boundaries. To facilitate this, we introduce two datasets, HMDB-AD and HMDB-Violence, to challenge models with diverse action-based anomalies. These datasets are derived from the HMDB51 action recognition dataset. We further present Multi-Frame Anomaly Detection (MFAD), a novel method built upon the AI-VAD framework. AI-VAD utilizes single-frame features such as pose estimation and deep image encoding, and two-frame features such as object velocity. They then apply a density estimation algorithm to compute anomaly scores. To address complex multi-frame anomalies, we add a deep video encoding features capturing long-range temporal dependencies, and logistic regression to enhance final score calculation. Experimental results confirm our assumptions, highlighting existing models limitations with new anomaly types. MFAD excels in both simple and complex anomaly detection scenarios.

## 1 Introduction

As the volume of recorded video content continues to grow, the need for robust and efficient video anomaly detection methods increases. The ability to automatically identify unusual events or behaviors within videos not only holds the promise of enhancing security but also offers the potential to reduce the manpower required for monitoring. However, achieving truly effective video anomaly detection remains a significant unsolved challenge, due to the diverse range of anomalies that can occur in real-world scenarios.

By nature, anomalous behaviors are rare. Thus, video anomaly detection (VAD) is often treated as a semi-supervised problem, where models are trained exclusively on normal videos and must subsequently distinguish between normal and abnormal videos during inference.

While current benchmark datasets vary in complexity, they share a common limitation in their narrow definition of anomalies. The three datasets, UCSD Ped2 (Mahadevan et al., 2010), CUHK Avenue (Lu et al., 2013), and ShanghaiTech Campus (Liu et al., 2018), tend to limit anomalies primarily to novel object detection (Ped2, ShanghaiTech) or simple movement anomalies (Avenue).

Recent advancements in video anomaly detection predominantly relied on analyzing a few frames or even individual frames in isolation. Researchers predominantly choose between two approaches: reconstruction-based and prediction-based methods. Reconstruction-based methods (Nguyen & Meunier, 2019; Luo et al., 2017; Gong et al., 2019; Fan et al., 2018; Hasan et al., 2016) typically employ auto-encoders to learn representations of normal frames, reconstructing them accurately, while anomalous frames result in a higher reconstruction error. Prediction-based methods (Liu et al., 2018; Lu et al., 2019; Yu et al., 2020; Park et al., 2020) focus on predicting the next frame from a sequence of consecutive frames.

These few-frame based methods achieved impressive results, surpassing an AUC score of 99% (Reiss & Hoshen, 2022; Liu et al., 2021) on Ped2, over 93% (Reiss & Hoshen, 2022) on Avenue, and

exceeding 85% (Reiss & Hoshen, 2022; Barbalau et al., 2023) on ShanghaiTech, the most complex of the benchmark datasets.

Without a shift in research focus and assumptions, the existing datasets, results, and recurring research patterns may suggest that the field of video anomaly detection is nearing a plateau.

This paper emphasizes the necessity of broadening the scope of what constitutes an anomaly. We propose two novel datasets specifically designed to assess the detection of complex action-based anomalies. These datasets, referred to as HMDB-AD and HMDB-Violence, build upon the HMDB51 action recognition dataset and define different actions as normal or abnormal activities. By analyzing the performance of various methods on our datasets, we underscore the limitations of existing approaches and advocate for further research on more comprehensive anomaly types.

Building upon the foundation laid by AI-VAD (Reiss & Hoshen, 2022), we introduce Multi-Frame Anomaly Detection (MFAD), a novel method aimed at achieving balanced performance, excelling in both simple and complex anomaly detection. AI-VAD utilizes a two-step approach: first, it extracts multiple features and then employs density estimation algorithms to calculate anomaly scores. In their work, they rely on single-frame features like deep image encoding (using a pretrained encoder) and human pose estimations, along with two-frame features such as object velocity. We extend this method by introducing deep video encoding features to capture multi-frame, long-range temporal relationships. MFAD adheres to the AI-VAD framework, computing final scores for each feature using a density estimation algorithm. Additionally, we incorporate logistic regression to enhance the relationships between different feature scores and achieve more accurate final scores.

We extensively evaluate our method on classic benchmark datasets as well as on our newly proposed datasets. The experiments validate the added value of both video encoding features and the logistic regression module. Our method achieves competitive results on Ped2, Avenue, and ShanghaiTech, and greatly outperforms recent methods on HMDB-AD and HMDB-Violence. As a result, it offers a more versatile video anomaly detection solution capable of detecting a broader range of anomalies across various scenarios.

Our key contributions are:

- We highlight the limitations of current video anomaly detection benchmarks and advocate for further research in general video anomaly detection.
- We present MFAD, a novel method capable of effectively handling both simple, few-frame anomalies and complex, multi-frame anomalies.
- We provide two datasets designed for assessing a model's performance on multi-frame action-based anomalies.

## 2 RELATED WORK

### 2.1 VIDEO ANOMALY DETECTION DATASETS

The datasets commonly used in video anomaly detection can be broadly categorized into two groups, reflecting the shift brought about by the advent of deep learning from approximately 2013 to 2018. Detailed comparison can be found in Table 1

Early datasets are notably smaller and often considered practically solved, include Subway Entrance (Adam et al., 2008), Subway Exit (Adam et al., 2008), UMN (of Minnesota, 2006), UCSD Ped1 (Mahadevan et al., 2010), UCSD Ped2 (Mahadevan et al., 2010), and CUHK Avenue (Lu et al., 2013). With the exception of UMN, these datasets feature only a single scene.

In contrast, more recent datasets have grown significantly in both scale and complexity. This newer group includes ShanghaiTech Campus (Liu et al., 2018), Street Scene (Ramachandra & Jones, 2020), IITB Corridor (Rodrigues et al., 2020), UBNormal (Acsintoae et al., 2022), and the most recent and largest of them all, NWPU Campus (Cao et al., 2023).

It is worth noting that among these datasets, only three have gained popularity as benchmarks: UCSD Ped2, CUHK Avenue, and ShanghaiTech Campus. However, as discussed in this paper, each of these benchmarks has its own set of limitations that motivate the need for further research in the field of video anomaly detection.

Other datasets that can be considered are UCF-Crime (Sultani et al., 2018) and XD-Violence (Wu et al., 2020). These datasets are built for fully supervised VAD learning and therefore are orders of magnitude larger than current benchmarks for unsupervised VAD such as this work. We follow previous studies and don't use them for our comparisons.

Table 1: Comparison of the different video anomaly detection datasets.

| Dataset | # Frames | | |
| --- | --- | --- | --- |
| | Total | Train | Test |
| Subway Entrance (Adam et al., 2008) | 86,535 | 18,000 | 68,535 |
| Subway Exit (Adam et al., 2008) | 38,940 | 4,500 | 34,440 |
| UMN (of Minnesota, 2006) | 7,741 | – | – |
| USCD Ped1 (Mahadevan et al., 2010) | 14,000 | 6,800 | 7,200 |
| USCD Ped2 (Mahadevan et al., 2010) | 4,560 | 2,550 | 2,010 |
| CUHK Avenue (Lu et al., 2013) | 30,652 | 15,328 | 15,324 |
| ShanghaiTech Campus (Liu et al., 2018) | 317,398 | 274,515 | 42,883 |
| Street Scene (Ramachandra & Jones, 2020) | 203,257 | 56,847 | 146,410 |
| IITB Corridor (Rodrigues et al., 2020) | 483,566 | 301,999 | 181,567 |
| UBnormal (Acsintoae et al., 2022) | 236,902 | 116,087 | 92,640 |
| NWPU Campus (Cao et al., 2023) | 1,466,073 | 1,082,014 | 384,059 |
| HMDB-AD (ours) | 92,585 | 58,790 | 33,795 |
| HMDB-Violence (ours) | 204,471 | 140,377 | 64,094 |

## 2.2 HMDB51 ACTION RECOGNITION DATASET

The HMDB51 (Kuehne et al., 2011) dataset, originally designed for action recognition (AR), is relatively small in scale. It is a collection of 6,766 video clips distributed across 51 distinct categories. Most other datasets are significantly larger and more diverse: SSv2 (Qualcomm, 2018), Kinetics-400 (Kay et al., 2017), Kinetics-600 (Carreira et al., 2018), Kinetics-700-2020 (Smaira et al., 2020) each consist of hundred of thousands of frames and hundreds of different classes. A comparison can be found in Table 2.

The HMDB51 dataset draws content from various sources, ensuring diversity. In this dataset, each class consists of no less than 101 video clips.

Table 2: Comparison of selected action recognition datasets.

| Dataset | Total Clips | # Classes |
| --- | --- | --- |
| UCF-101 (Soomro et al., 2012) | 13,320 | 101 |
| ActivityNet-200 (Caba Heilbron et al., 2015) | 28,108 | 200 |
| Something-Something v1 (Goyal et al., 2017) | 108,499 | 174 |
| Something-Something v2 (Qualcomm, 2018) | 220,847 | 174 |
| Kinetics-400 (Kay et al., 2017) | 306,245 | 400 |
| Kinetics-600 (Carreira et al., 2018) | 495,547 | 600 |
| Kinetics-700-2020 (Smaira et al., 2020) | 647,907 | 700 |
| HMDB51 (Kuehne et al., 2011) | 6,766 | 51 |

## 2.3 VIDEO ANOMALY DETECTION METHODS

**Hand-crafted feature based methods** Numerous methods, spanning both classical and contemporary approaches, adhere to a two-stage anomaly detection framework. This framework involves an initial step of extracting hand-crafted features, specifically selected by the researcher and not learned through a deep neural network model. Subsequently, another algorithm is applied to compute anomaly scores.

Early techniques used classic image and video features, including the histogram of oriented optical flow (HOF) (Chaudhry et al., 2009; Perš et al., 2010; Colque et al., 2017; Sabzalian et al., 2019), histogram of oriented gradients (HOG) (Sabzalian et al., 2019), and SIFT descriptors (Lowe, 2004). In more recent developments, the proliferation of deep learning has facilitated the adoption of off-the-shelf models, such as object detectors, for feature extraction. For instance, in the case of AI-VAD (Reiss & Hoshen, 2022), a combination of pose estimations, optical flow predictions, object detection, and deep image encodings is used to construct robust feature representations.

Following feature extraction, classical methodologies often employed scoring techniques such as density estimation algorithms (Latecki et al., 2007; Glodek et al., 2013; Eskin et al., 2002). Recent approaches have demonstrated the effectiveness of integrating these features with another learning model (Liu et al., 2021).

**Reconstruction and prediction based methods**  In recent years, the increasing prominence of deep learning has driven the widespread adoption of both reconstruction and prediction based methods in video anomaly detection.

Reconstruction-based (Nguyen & Meunier, 2019; Luo et al., 2017; Gong et al., 2019; Fan et al., 2018; Hasan et al., 2016) approaches often utilize auto-encoders to learn representations of normal video frames and subsequently detect abnormal frames by identifying higher reconstruction errors. However, the powerful generalization ability of modern auto-encoders can often also reconstruct anomalies. Thus, making it harder to differentiate normal and abnormal frames.

Prediction-based (Liu et al., 2018; Lu et al., 2019; Yu et al., 2020; Park et al., 2020) models forecast the subsequent frame by leveraging a sequence of preceding frames, employing time sensitive architectures such as LSTMs, memory networks, 3D auto-encoders and transformers. This predictive approach often yields superior results compared to similar reconstruction-based techniques (Park et al., 2020), as it captures more complex forms of anomalies. Nevertheless, with the minimal differences between consecutive video frames, these methods face similar challenges to reconstruction-based approaches in respect to modern generators.

**Auxiliary tasks methods**  Expanding beyond reconstruction and prediction, some models incorporate diverse self-supervised auxiliary tasks, with task success determining frame anomaly scores. These tasks include jigsaw puzzles (Wang et al., 2022), time direction detection (Wei et al., 2018), rotation prediction (Gidaris et al., 2018), and more. Barbalau et al. (2023); Georgescu et al. (2021), train a single deep backbone on multiple self-supervised tasks and achieve state-of-the-art results on the benchmark datasets.

## 3 PROPOSED DATASETS

We introduce two novel datasets designed to assess the capability of various models in detecting forms of anomalies not covered by existing benchmarks. These datasets emphasize action-based anomalies, a category absent in current benchmarks. The first dataset, referred to as HMDB-AD, aligns with the conventional definition of normal activities (walking and running) but challenges models with abnormal behaviors that demand a broader context for detection (climbing and performing a cartwheel). In contrast, the larger and more intricate HMDB-Violence dataset divides 16 action categories into 7 violent (abnormal) and 9 non-violent (normal) activities. This categorization necessitates models to consider a wide range of behaviors when classifying events as either normal or abnormal, making it a closer representation of real-world scenarios. A comparison to other video anomaly detection datasets can be found in Table 1.

**HMDB-AD dataset**  HMDB-AD is the simpler dataset among the two introduced in this paper. It consists of 995 video clips, divided into 680 training videos and 315 testing videos. Normal activities within this dataset are running and walking, aligning with their respective HMDB51 classes. Abnormal activities are climbing and performing a cartwheel. The training dataset contains only of normal videos: 207 running videos and 473 walking videos. Meanwhile, the test dataset has both abnormal videos and randomly selected normal videos; 107 cartwheel videos, 108 climbing videos, 25 running videos, and 75 walking videos. Frames from the videos can be viewed in Appendix A.1.

**HMDB-Violence dataset**  HMDB-Violence is the larger and more complex of the two datasets presented in this paper. It has 2,566 videos, with a distribution of 1,601 training videos and 965 testing videos. The train set has nine normal categories: running (221 videos), walking (517), waving (98), climbing (104), hugging (110), throwing (96), sitting (134), turning (222), and performing a cartwheel (99). In the test set, there are seven abnormal categories: falling (136), fencing (116), hitting (127), punching (126), using a sword (127), shooting (103), and kicking (130). Additionally, the test set includes 100 videos randomly sampled from the various normal categories: turning (18), walking (31), running (11), sitting (8), hugging (8), performing a cartwheel (8), climbing (4), throwing (6), and waving (6). The abnormal activities in HMDB-Violence are characterized by their violent nature. Examples can be viewed in Appendix A.2.

**Annotations**  We maintain a consistent labeling for every frame within a video. If a video represents a normal action category, all its frames are labeled as normal. Conversely, if it belongs to an abnormal action category, all frames are marked as as abnormal. This simple labeling approach works, as the actions within these videos effectively occupy the entire duration, leaving minimal room for unrelated "spare" frames.

## 4 MFAD: MULTI-FRAME ANOMALY DETECTION

Our method, MFAD, consists of three key stages: feature extraction, per-feature score computation, and logistic regression. We extract four types of features: object velocities, human pose estimations, deep image encodings, and deep video encodings. For each of these features, we independently calculate density scores. We then employ a logistic regression model to optimally fuse the scores across these four feature kinds. Lastly, we smooth, Gaussian, to produce the final anomaly scores.

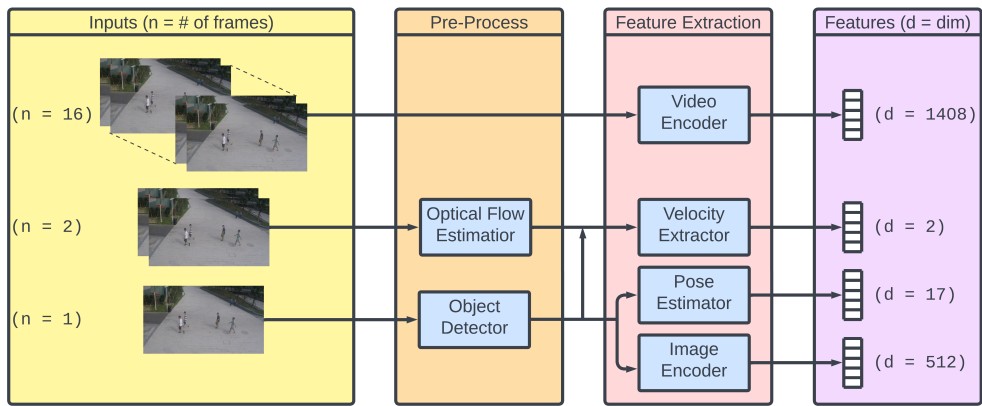

Figure 1: An overview of our feature extraction process.

### 4.1 FEATURE EXTRACTION

**Few-Frame Features**  In line with Reiss & Hoshen (2022); Liu et al. (2021), we extract object bounding boxes and optical flows from each frame. We then extract human pose estimations, object velocities, and deep image encodings. These features are derived from individual frames (pose and image encoding) or pairs of frames (velocity) enabling the detection of straightforward anomalies such as novel objects.

**Multi-Frame Features**  Recognizing the necessity for detecting complex anomalies that span multiple frames, we introduce a deep video encoder. This encoder captures features in a manner similar to deep image encoding but takes into account longer frame sequences (in our case, 16 frames). For this purpose, we leverage VideoMAEv2 (Wang et al., 2023), a state of the art video foundation model. Subsequently, we process these features in a fashion similar to AI-VAD (Reiss & Hoshen, 2022).

## 4.2 DENSITY SCORE CALCULATION

We employ a Gaussian Mixture Model (GMM) for the two-dimensional velocity features and the k-nearest neighbors (kNN) algorithm for the high-dimensional pose, image encoding, and video encoding features. Subsequently, we compute the minimum and maximum density scores for the training set and use them to calibrate the test scores during inference.

**Max Feature**  We add a fifth feature, denoted as $\max \in [0,1]^{\#frames}$. After calculating the density scores per feature, we aggregate them to a new feature that holds the maximum feature score per frame.

$$\max = \max\{\text{pose score, velocity score, image encoding score, video encoding score}\}$$

Our experiments show the added value of this feature.

## 4.3 LOGISTIC REGRESSION

In order to improve the accuracy of our final anomaly score computation, we incorporate logistic regression as the final step of our method. In this setup, we denote $X \in [0,1]^{\#frames \times \#features}$ as the feature matrix and $y \in \{0,1\}^{\#frames}$ as our ground truth labels. Our final scores are:

$$h_\theta(X) = \sigma(WX + B)$$

where $\sigma(t) = \frac{1}{1+e^{-t}}$ is the sigmoid function and $\theta = (W, B)$ are the parameters we want to optimize. Our loss function is:

$$L(h_\theta(X), y) = -y \log(h_\theta(X)) - (1 - y) \log(1 - h_\theta(X))$$

During its training phase, we randomly sample a small fraction of the test frames for model training, while the remainder is used for evaluation. It is crucial to emphasize that the frames utilized for training are excluded from the evaluation process for our reported results, ensuring the validity of our findings.

The final step in our method is applying Gaussian smoothing to the anomaly scores.

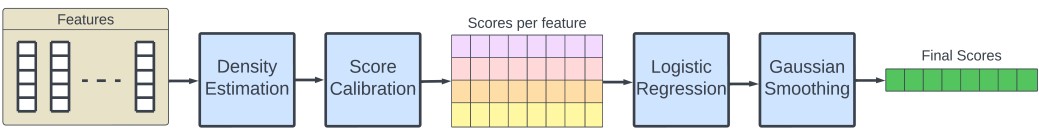

Figure 2: An overview of our anomaly score calculation during inference.

# 5 EXPERIMENTS

## 5.1 DATASETS

In addition to HMDB-AD and HMDB-Violence, we evaluate MFAD on the three benchmark video anomaly detection datasets: UCSD Ped2, CUHK Avenue, and ShanghaiTech Campus. These datasets are primarily outdoor surveillance camera footage, with the sole normal activity being pedestrian movement. Details are presented in Table 1.

**UCSD Ped2**  The UCSD Ped2 dataset has 16 training videos and 12 testing videos, all situated within a single scene. Abnormal events in this dataset include the appearance of skateboards, bicycles, or cars within the video frame. Videos are standardized to a resolution of $240 \times 360$ pixels.

**CUHK Avenue**  The CUHK Avenue dataset has 16 training videos and 21 testing videos, all within a single scene. Anomalies within this dataset are activities such as running, throwing objects, and bike riding. All videos have a resolution of $360 \times 640$ pixels.

**ShanghaiTech Campus**  ShanghaiTech Campus stands as the largest and most complex dataset among the three, featuring 330 training videos and 107 testing videos distributed across 13 distinct scenes. Notably, two of these scenes involve non-stationary cameras, resulting in varying angles between videos of the same scene. Abnormal events primarily include running and the presence of cars and bikes. All videos have a resolution of $480 \times 856$ pixels.

## 5.2 IMPLEMENTATION DETAILS

We adopt the code from AI-VAD for extracting velocity, pose, and deep image encoding features. For our new deep video features, we leverage the state-of-the-art video foundation model, Video-MAEv2 (Wang et al., 2023), with the publicly available pretrained weights, fine-tuned on the SSv2 dataset (*vit_g_hybrid_pt_1200e_ssv2_ft*). Our encoding process is carried out on non-overlapping consecutive blocks of 16 frames, extracting Temporal Action Detection (TAD) features for each block. In our experiments we found no difference in results between non-overlapping blocks and sliding-window blocks. When employing the nearest neighbors algorithm to the video encoding features, we set $k = 1$.

We employ AlphaPose for pose estimation, derive object velocity through optical flows computed via FlowNet2, and utilize YOLOv3 for object detection.For deep image encoding, we leverage CLIP, using a ViT-32 backbone.

## 5.3 ANOMALY DETECTION RESULTS

Our results are based on our optimal model configuration, see Section 5.4. This configuration involves leveraging all four feature types and the `max` feature, while training a logistic regression model on a random 2% of the test set frames for computing final anomaly scores. It is crucial to note that the data used for training the logistic regression model is not included in the evaluation. To ensure reliability, we repeat this final step 100 times and report the mean AUC result along with the standard deviation. The consistently low standard deviation across all datasets underscores the stability of our method.

MFAD demonstrates competitive results on the well-established benchmark datasets, with modest differences of approximately -0.3%, -0.8%, and -1.1% from the state-of-the-art results on Ped2, Avenue, and ShanghaiTech, respectively. The true strength of our approach becomes evident when applied to the newly introduced datasets, HMDB-AD and HMDB-Violence. On these datasets, we achieve substantial improvements of 19.9% and 9.7%, respectively. MFAD was tested against four different methods on these new datasets, including AI-VAD (Reiss & Hoshen, 2022), upon which our work is built and is the state-of-the-art on the ShanghaiTech dataset. This substantial enhancement highlights the generalizability of our approach to various complex anomalies, without majorly impacting our detection ability of simple anomalies, underscoring the significance of our contributions. For detailed comparison see Table 3.We further report the configuration of MFAD without image encoding (IE) features, improving results on ShanghaiTech by 0.2%.

MFAD faces similar challenges to previous methods when evaluated against the benchmark datasets. Particularly, the object-oriented aspect of MFAD struggles when confronted with scenarios involving closely clustered pedestrians.

In addition to quantitative evaluations, we conducted qualitative analyses on videos from the ShanghaiTech dataset, which feature more complex anomalies beyond novel object detection. These anomalies are shown in Appendix B. The positive impact of our method is clearly evident in Fig. 3, where abnormal frames receive higher anomaly scores, while normal frames receive lower anomaly scores, further validating our method.

## 5.4 ABLATION STUDY

We perform an ablation study to determine two factors: the added benefit of the video encoding feature, and the most favorable configuration for the logistic regression module.

**Feature Selection**  In their ablation study, Reiss & Hoshen (2022) demonstrated the incremental value of their three distinct feature types: pose estimation, deep image encoding, and velocity fea-

Table 3: Comparison to frame-level AUC. Best (bold), second (underlined) and third (italic) best results are highlighted. Image encoding features is denoted by IE.

| Method | Ped2 | Avenue | ShanghaiTech | HMDB-AD | HMDB-Violence |
|---|---|---|---|---|---|
| HF2-VAD (Liu et al., 2021) | **99.3%** | 91.1% | 76.2% | – | – |
| AED (Georgescu et al., 2022) | 98.7% | 92.3% | 82.7% | – | – |
| HSC-VAD (Sun & Gong, 2023) | 98.1% | **93.7%** | 83.4% | – | – |
| DLAN-AC (Yang et al., 2022) | 97.6% | 89.9% | 74.7% | – | – |
| SSMTL (Georgescu et al., 2021) | 97.5% | 91.5% | 82.4% | – | – |
| LBR-SPR (Yu et al., 2022) | 97.2% | 90.7% | 72.6% | – | – |
| AMMCNet (Cai et al., 2021) | 96.6% | 86.6% | 73.7% | – | – |
| AI-VAD (Reiss & Hoshen, 2022) | 99.1% | 93.3% | **85.9%** | *70.1%* | *70.5%* |
| Jigsaw Puzzles (Wang et al., 2022) | 99.0% | 92.2% | 84.3% | 53.8% | 52.7% |
| MNAD (Park et al., 2020) | 97.0% | 88.5% | 70.5% | 56.3% | 51.3% |
| MPN (Lv et al., 2021) | 96.9% | 89.5% | 73.8% | 58.8% | 53.7% |
| MFAD (Ours) | *99.0% ± 0.5%* | *92.9% ± 0.5%* | *84.8% ± 0.4%* | **90.0% ± 0.4%** | **80.2% ± 0.2%** |
| MFAD w/o IE (Ours) | 98.4% ± 0.7% | 90.7% ± 0.5% | 85.0% ± 0.4% | 86.9% ± 0.5% | 76.0% ± 0.2% |

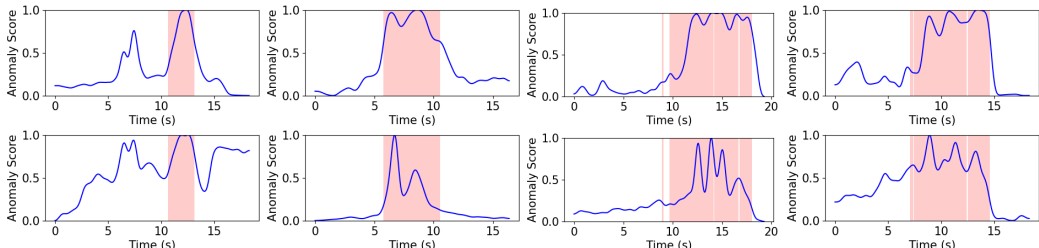

Figure 3: Qualitative results from three ShanghaiTech videos: 01_0028 (left), 03_0032 ($2^{nd}$ left), 03_0039 ($2^{nd}$ right), 07_0008 (right). In each pair, MFAD (top) is compared to AI-VAD (Reiss & Hoshen, 2022) (bottom). Anomalous sections are highlighted in red, while the anomaly scores, ranging from 0 to 1, are the blue line. These videos feature complex, behavior-based anomalies rather than novel object detection scenarios, that are more common in this dataset. Clearly, MFAD improves both detecting anomalies and accurately assessing normal parts of the video. Best viewed in color.

tures, as well as the added effect of Gaussian smoothing. Expanding upon their work, we test the impact of incorporating deep video encoding features in different forms. Our study consists of tests involving both video encoding features in isolation, the combination of all four feature types, and the addition of an extra max feature, that has the value of the maximum feature score per each frame. Furthermore, we explore the substitution of image encoding features with video encoding features due to their semantic similarity.

As presented in Table 4, utilizing solely video encoding features yields impressive performance for multi-frame anomalies. However, this specialization comes at the cost of lower performance on the traditional video anomaly detection datasets. On the other hand, employing all four feature types results in a comprehensive and well-balanced model that performs admirably across all datasets, even though it may not achieve the top rank in any specific dataset.

**Logistic Regression** When incorporating the logistic regression model, we conducted experiments to assess the impact of varying amounts of additional training data extracted from the testing set. Specifically, we explored using 1-5%, 10%, 20%, 50%, 90% of the frames for the training. We used both the configuration using only the four basic features and the configuration also using the additional max feature. The results, as presented in Table 5, indicate that the amount of extra training data has minimal effects, as long as there is some extra data. We repeated each configuration 100 times and reported both the mean and standard deviation values. The consistently low standard deviation values observed across all configurations and datasets underscore the robustness of our approach. We chose 2% extra data with the max feature as the optimal trade-off between extra data and efficacy.

Table 4: Comparison of different model configurations, evaluating the impact of various feature types, including pose features (P), velocity features (V), image encoding features (IE), and video encoding features (VE), on the model's performance. max is the max value between {P, V, IE, VE}. Best and second best results are in bold and underlined, respectively.

| Configuration | Ped2 | Avenue | ShanghaiTech | HMDB-AD | HMDB-Violence |
|---|---|---|---|---|---|
| VE | 80.3% | 87.9% | 71.3% | 84.9% | 75.8% |
| P + V (Reiss & Hoshen, 2022) | 98.7% | 86.8% | **85.9%** | 54.2% | 56.1% |
| P + V + IE (Reiss & Hoshen, 2022) | **99.1%** | **93.5%** | 85.1% | 71.2% | 67.9% |
| P + V + VE | 95.8% | 91.0% | 83.5% | 77.8% | 70.3% |
| P + V + IE + VE | 96.8% | 92.6% | 83.0% | 82.9% | 75.2% |
| P + V + IE + VE + max | 97.0% | 92.8% | 82.1% | **85.1%** | **76.7%** |

Table 5: Performance comparison between various model configurations, with different amounts of training data for the logistic regression model. $\alpha$ represents the proportion of test set frames employed for the training, with these frames excluded from model evaluation. We repeat the process 100 times, and both mean and standard deviation values are reported. The first half uses the basic four features, and the second half also uses the extra max feature. Best and second best results highlighted in bold and underlined, respectively. The minimal difference in results between different values of $\alpha > 0\%$ is evident.

| Configuration | Ped2 | Avenue | ShanghaiTech | HMDB-AD | HMDB-Violence |
|---|---|---|---|---|---|
| $\alpha = 0\%$ | 96.8% | 92.6% | 83.0% | 82.9% | 75.2% |
| $\alpha = 1\%$ | 98.5% ± 1.1% | 92.5% ± 0.6% | 84.5% ± 0.6% | 89.6% ± 0.6% | 79.6% ± 0.3% |
| $\alpha = 2\%$ | 99.0% ± 0.6% | 92.7% ± 0.7% | 84.7% ± 0.4% | 89.9% ± 0.4% | 79.7% ± 0.2% |
| $\alpha = 3\%$ | 99.2% ± 0.5% | 92.7% ± 0.7% | 84.8% ± 0.4% | 89.9% ± 0.3% | 79.7% ± 0.1% |
| $\alpha = 4\%$ | 99.4% ± 0.3% | 92.7% ± 0.7% | 84.7% ± 0.3% | 89.9% ± 0.3% | 79.8% ± 0.1% |
| $\alpha = 5\%$ | 99.4% ± 0.4% | 93.0% ± 0.6% | 84.8% ± 0.3% | 90.0% ± 0.2% | 79.8% ± 0.1% |
| $\alpha = 10\%$ | 99.5% ± 0.2% | 92.9% ± 0.6% | 84.8% ± 0.2% | 90.1% ± 0.2% | 79.8% ± 0.1% |
| $\alpha = 20\%$ | 99.6% ± 0.1% | 93.1% ± 0.5% | 84.8% ± 0.2% | 90.2% ± 0.2% | 79.8% ± 0.1% |
| $\alpha = 50\%$ | **99.7% ± 0.1%** | 93.1% ± 0.3% | 84.8% ± 0.2% | 90.2% ± 0.2% | 79.7% ± 0.3% |
| $\alpha = 90\%$ | 99.7% ± 0.3% | **93.2% ± 0.6%** | 84.8% ± 0.7% | 90.2% ± 0.5% | 79.8% ± 0.8% |
| $\alpha = 0\% + \text{max}$ | 97.0% | 92.8% | 82.1% | 85.1% | 76.7% |
| $\alpha = 1\% + \text{max}$ | 98.5% ± 0.8% | 92.5% ± 0.6% | 84.5% ± 0.6% | 89.8% ± 0.5% | 80.2% ± 0.3% |
| $\alpha = 2\% + \text{max}$ | 99.0% ± 0.5% | 92.9% ± 0.5% | 84.8% ± 0.4% | 90.0% ± 0.4% | **80.2% ± 0.2%** |
| $\alpha = 3\% + \text{max}$ | 99.1% ± 0.7% | 92.9% ± 0.6% | 85.0% ± 0.3% | 90.0% ± 0.3% | **80.2% ± 0.2%** |
| $\alpha = 4\% + \text{max}$ | 99.3% ± 0.5% | 93.0% ± 0.5% | 85.1% ± 0.3% | 90.1% ± 0.3% | **80.2% ± 0.2%** |
| $\alpha = 5\% + \text{max}$ | 99.3% ± 0.4% | 93.0% ± 0.4% | 85.1% ± 0.3% | 90.2% ± 0.3% | **80.2% ± 0.2%** |
| $\alpha = 10\% + \text{max}$ | 99.5% ± 0.2% | 93.0% ± 0.4% | 85.2% ± 0.2% | 90.2% ± 0.2% | **80.2% ± 0.2%** |
| $\alpha = 20\% + \text{max}$ | 99.6% ± 0.1% | 93.0% ± 0.3% | 85.2% ± 0.2% | 90.3% ± 0.2% | 80.1% ± 0.2% |
| $\alpha = 50\% + \text{max}$ | **99.7% ± 0.1%** | 93.0% ± 0.3% | **85.3% ± 0.2%** | **90.4% ± 0.2%** | 80.1% ± 0.3% |
| $\alpha = 90\% + \text{max}$ | 99.7% ± 0.2% | 93.1% ± 0.6% | 85.3% ± 0.6% | 90.4% ± 0.5% | 80.1% ± 0.8% |

## 6 CONCLUSION

Our paper introduces a broader interpretation of anomalies, encompassing both simple anomalies, commonly found in existing benchmarks, and multi-frame complex anomalies. Building upon the foundation laid by AI-VAD (Reiss & Hoshen, 2022), we present a novel method that achieves state-of-the-art performance on our proposed datasets while remaining competitive with recent methods on benchmark datasets. We introduce two new datasets of varying complexity, designed to assess the ability of future models to detect complex action-based anomalies.

In future work, we aim to explore even more intricate types of anomalies, such as location and time-based anomalies (e.g. detecting normal actions occurring at abnormal locations or times) thus further advancing the field of general anomaly detection in videos.

**Ethics Statement**   We have carefully reviewed the ICLR Code of Ethics and have taken steps to ensure that our work aligns with its principles. All the data and pre-trained models employed in our experiments are publicly accessible and pose no ethical concerns.

**Reproducibility Statement**   To ensure reproducibility of our results, we provide a comprehensive reproduction guide and all of our code in the supplementary material.

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

## A  SAMPLES FROM PROPOSED DATASETS

### A.1  HMDB-AD EXAMPLES

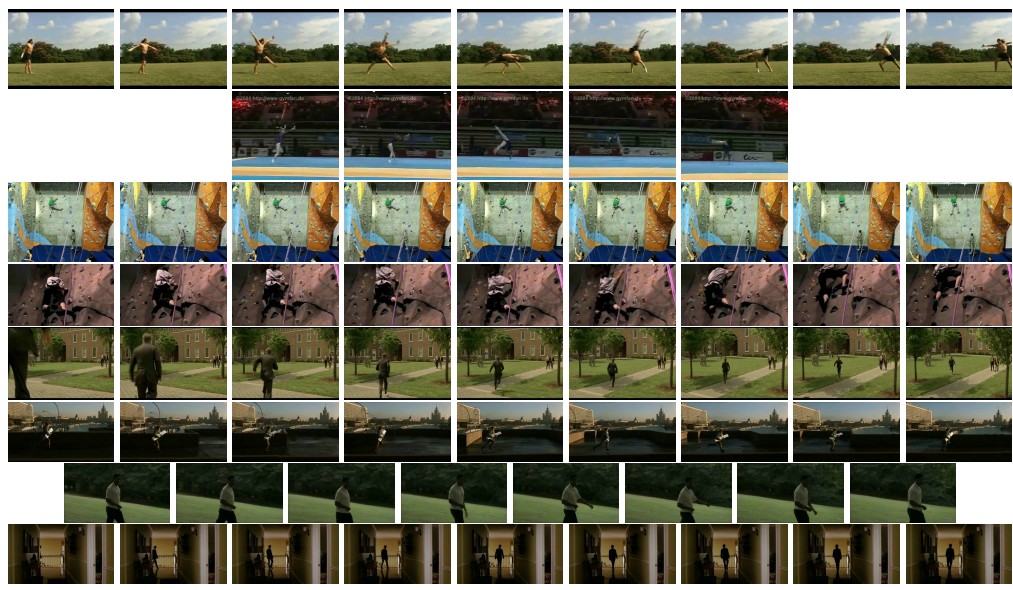

Figure 4: HMDB-AD examples: cartwheel (2), climb (2), run (2), walk (2).

### A.2  HMDB-VIOLENCE EXAMPLES

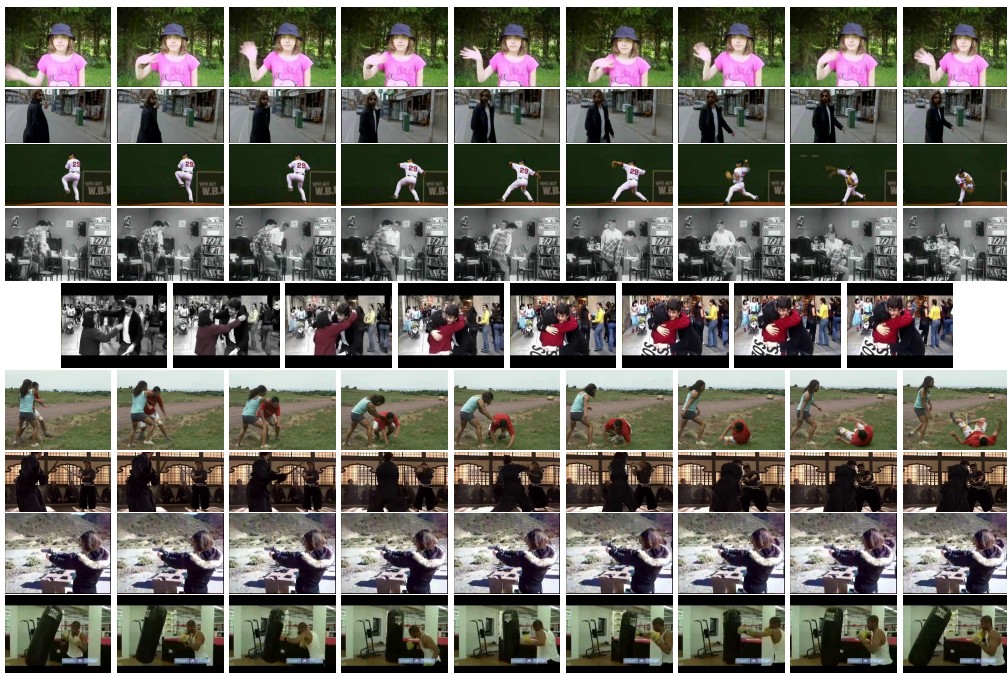

Figure 5: HMDB-Violence examples: wave, turn, throw, sit, hug, fall, sword, shoot, punch.

## B VIDEOS FOR QUALITATIVE ANALYSES

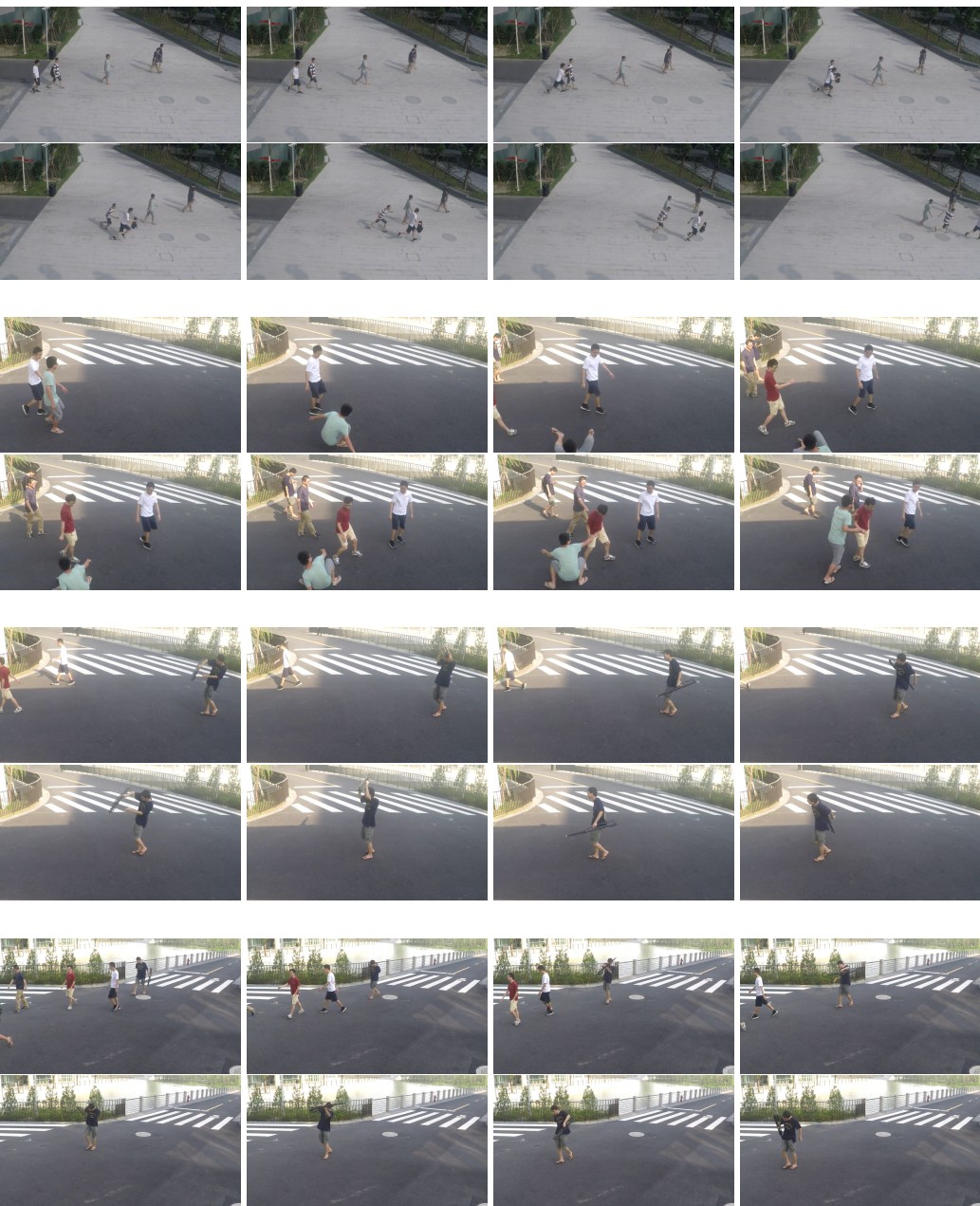

Figure 6: The anomalies from 01_0028, 03_0032, 03_0039, 07_0008 (top to bottom, respectively) videos from ShanghaiTech Campus dataset.

