# OpenReview forum: "Beyond the Benchmark: Detecting Diverse Anomalies in Videos"
_ICLR.cc/2024/Conference — Submitted to ICLR 2024_

### Official Review · Reviewer_fVMW · 2023-10-15

**Soundness:** 3 good
**Presentation:** 3 good
**Contribution:** 1 poor
**Rating:** 3
**Confidence:** 5

**Summary:**

The authors study anomaly detection in video, presenting two datasets with video-level annotations and an adapted version of AI-VAD, called MFAD, which performs well on the proposed datasets. The presented method is also evaluated on three existing datasets, being compared with AI-VAD and other methods from literature.

**Strengths:**

- Anomaly detection is an interesting and timely topic.
- The paper is well written and easy to follow.

**Weaknesses:**

- The proposed method is incremental w.r.t. AI-VAD.
- UCF-Crime and XD-Violence datasets are not included in the comparison provided in Table 1.
- The proposed datasets are rearranged subsets of HMDB51. There was no manual annotation involved, or at least, the authors did not mentione anything about it. Therefore, it is hard to consider the proposed datasets as entirely new. Just as the method, this contribution is incremental.
- The comparison in Table 1 does not reflect the difficulty / diversity advantages suggested by the authors. I do not see the benefits of the proposed datasets w.r.t. recent benchmarks such as UCF-Crime, XD-Violence or UBnormal.
- The proposed datasets contain video-level annotations (the videos are labeled either as normal or abnormal), while other benchmarks contain frame or pixel annotations. I believe this type of annotation does not reflect a realistic scenario.
- There is no time evaluation reported for the presented method. Anomaly detection methods are expected to run in real-time, but it is not clear if MFAD can do this. To me, the method is a bit heavy.
- There are some some language corrections to be made, e.g.:
  - "It’s crucial to" => "It is crucial to" (language abbrevations should be avoided in formal language).

**Questions:**

Please see the weaknesses.

---

> ### Author Response · Authors · 2023-11-20
>
> Thank you for your insightful feedback, and for your dedication to improving our research.
>
>
> Our contribution on top of AI-VAD:
>
> Our paper's core contribution lies in addressing the underexplored realm of diverse multi-frame anomalies, a facet largely unaddressed in prior research. While drawing inspiration from AI-VAD, our work crucially emphasizes the comparison between prevalent benchmark datasets, primarily featuring simple anomalies, and our novel datasets, introducing intricate, action-based anomalies. This contrast aims to encourage a broader perspective on anomalies, reflecting real-world scenarios. The significance of this comparison is evident in Table 3, which demonstrates a major improvement, of 19.9% and 9.7%, when using our method on the proposed datasets, while exposing limitations in the existing approaches.
>
>
> Time complexity & Real-Time constraints:
>
> Our method's multi-stage process and diverse feature extraction are aligned with the state-of-the-art Video Anomaly Detection (VAD) methods [1]. The reliance on established models ensures that our method is comparable in run time to various prior approaches [2, 3].
>
> We acknowledge that real-time application remains a challenge in current VAD research. As part of our future endeavors, we're exploring strategies to enhance computational efficiency. Our approach's modular structure enables parallel computation of individual features, offering a pathway to expedite processing. Furthermore, it provides our system an option for easily integrating future improved components that can speed up our method even more.
>
>
> UCF-Crime, XD-Violence or UBnormal:
>
> Though indeed used for VAD, UCF-Crime, XD-Violence and UBnormal are built for supervised VAD learning which is less compatible with the popular use case for Ped2, Avenue and ShanghaiTech Campus which is unsupervised VAD.
>
> While UCF-Crime, XD-Violence, and UBnormal datasets have been employed in VAD contexts, they are primarily tailored for supervised VAD learning. This framework contrasts with the prevalent use case for Ped2, Avenue, and ShanghaiTech Campus datasets, which predominantly function in unsupervised VAD scenarios.
>
> Considering the aforementioned time-complexity issue, adapting UCF-Crime and XD-Violence datasets for unsupervised VAD poses a challenge, due to the severe increase in their size compared to current benchmark datasets. UCF-Crime has a total of 128 hours of data and XD-Violence has around 217 hours, both orders of magnitude above ShanghaiTech which has 3.5 hours of data, Avenue with around 20 minutes and Ped2 with less than 5 minutes. Our smaller-sized HMDB-AD (around one hour of data) and HMDB-Violence (less than 2.5 hours) datasets fulfill a unique role in advancing VAD research, specifically addressing complex anomalies and keeping realistic processing run times.
>
> Following your interest on this topic we also added a short explanation to section 2.1 for future readers.
>
>
> Language corrections:
>
> We revised our paper with the language corrections you rightfully pointed out.
>
>
> We hope that we've sufficiently addressed your concerns and look forward to further fruitful discussions.
>
>
> [1] Attribute-based Representations for Accurate and Interpretable Video Anomaly Detection
>
> [2] SSMTL++: Revisiting self-supervised multi-task learning for video anomaly detection
>
> [3] A Hybrid Video Anomaly Detection Framework via Memory-Augmented Flow Reconstruction and Flow-Guided Frame Prediction

---

> > ### Comment · Reviewer_fVMW · 2023-11-22
> > **Reply to rebuttal**
> >
> > - _Time complexity._ It would have been better to report the FPS of the employed method to address the raised issue. This is still not clear.
> >
> > - _UCF-Crime, XD-Violence or UBnormal._ While these datasets do include anomalies at training time, there is nothing to stop us from discarding the labels or the training time anomalies and use these datasets in an unsupervised scenario.
> >
> > - Several other points, e.g. the novelty of the dataset, were not addressed by the authors. In summary, I think that the study is not well justified and has limited novelty.

---

### Official Review · Reviewer_A5Te · 2023-10-30

**Soundness:** 3 good
**Presentation:** 3 good
**Contribution:** 2 fair
**Rating:** 5
**Confidence:** 5

**Summary:**

This paper proposes a multi-frame-based video anomaly detection method, that builds on top of [1]. In [1] mostly frame-level attributes are included, while the proposed method extends the method of [1] by including multi-frames encoding features extracted across 16 frames.

Furthermore, the paper cherry picks two groups of anomalies from HMDB51 to show the importance of anomalies across the temporal axis.

The paper reports interesting results not only on the above two subsets of HMDB51, but also on few benchmark anomaly detection datasets. The results on the benchmark datasets are competitive compared to the chosen state-of-the-art methods, but outperforming them on the two subsets of HMDB51.

[1] Tal Reiss and Yedid Hoshen. Attribute-based Representations for Accurate and Interpretable
Video Anomaly Detection, December 2022. URL http://arxiv.org/abs/2212.00789.
arXiv:2212.00789 [cs].

**Strengths:**

-Good overview of existing methods and datasets used in anomaly detection.
-"Introducing" new videos to video anomaly detection for benchmarking.
-Comprehensive and competitive results on the public benchmarks and significantly higher results compared to state-of-the-art on the two subsets of HMDB51.
-Proper ablation study. which also shows the effect of video encoding features in Table 4.

**Weaknesses:**

Despite the interesting results, the paper's method sounds like a simple extension of [1] by introducing temporal features to [1].
Though the paper has cherry picked videos from HMDB51 and suggests using them for anomaly detection, they need claim this as their data (Table 1), which is not correct.
The abolition study shows that video encoder features alone are producing almost similar results with the entire set of features on the subsets of HMDB51, so what is the point in inclusion of other features?
What about including/cherry picking some other videos from HMDB51 that are normal, but are similar to the abnormal videos already included in the two subsets?

**Questions:**

Please see the previous section

---

> ### Author Response · Authors · 2023-11-20
>
> We thank you for the thorough review and for taking the time to consider our work and for your valuable comments. We provide answers to your questions and address your concerns below.
>
>
> Our contribution on top of AI-VAD:
>
> Our paper's core contribution lies in addressing the underexplored realm of diverse multi-frame anomalies, a facet largely unaddressed in prior research. While drawing inspiration from AI-VAD, our work crucially emphasizes the comparison between prevalent benchmark datasets, primarily featuring simple anomalies, and our novel datasets, introducing intricate, action-based anomalies. This contrast aims to encourage a broader perspective on anomalies, reflecting real-world scenarios. The significance of this comparison is evident in Table 3, which demonstrates a major improvement, of 19.9% and 9.7%, when using our method on the proposed datasets, while exposing limitations in the existing approaches.
>
>
> As per using only video encoding features:
>
> Our method's novelty relies on its capacity to effectively handle a broad range of anomalies. The ablation study in Section 5.4 illuminates that sole reliance on video encoding features notably deteriorates performance on simple anomalies prevalent in benchmark datasets. However, the combination of diverse feature types enabled our method to maintain comparable performance on simpler anomaly sets while notably enhancing results on complex anomalies within our proposed datasets. This versatility is crucial as it ensures the method's consistency across different anomaly types and datasets.
>
>
> Regarding adding normal videos to the test sets of proposed datasets:
>
> This is a very interesting suggestion. While not explicitly incorporating this, we note in Section 3 the inclusion of normal videos from each dataset's existing normal categories for testing, without using them for training. Although the exact implementation of testing similar yet untrained categories wasn’t carried out, our methodology accounts for this aspect through the inclusion of diverse normal videos in our testing.
>
>
> We highly value your feedback and hope that our responses have addressed your concerns.

---

### Official Review · Reviewer_HVoz · 2023-10-30

**Soundness:** 1 poor
**Presentation:** 1 poor
**Contribution:** 2 fair
**Rating:** 3
**Confidence:** 5

**Summary:**

Briefly summarize the paper and its contributions. This is not the place to critique the paper; the authors should generally agree with a well-written summary.
This paper proposes a method for video anomaly detection that goes beyond the limitations of current benchmark datasets. The authors introduce two new datasets, HMDB-AD and HMDB-Violence, which challenge models with diverse action-based anomalies. They also present a novel method called Multi-Frame Anomaly Detection (MFAD) that incorporates deep video encoding features to capture long-range temporal dependencies and logistic regression to enhance the final score calculation. The experimental results show that MFAD outperforms existing methods on both simple and complex anomaly detection scenarios.

**Strengths:**

(1) The paper addresses the limitation of current benchmark datasets for video anomaly detection and proposes two new datasets that allow for the detection of complex action-based anomalies. This expands the scope of what constitutes an anomaly and encourages further research on more comprehensive anomaly types.
(2) The proposed method, MFAD, simply incorporates deep video encoding features and logistic regression to effectively detect both simple and complex anomalies. The experimental results demonstrate the effectiveness of the method on benchmark datasets as well as the newly introduced datasets.
(3) The paper is well-structured and clear. Based on the two datasets proposed in the paper, the method used performs better than existing methods.

**Weaknesses:**

(1) The paper lacks a more detailed description of the datasets HMDB-AD and HMDB-Violence. It would be beneficial to provide more information on the distribution of normal and abnormal activities, and any specific challenges or characteristics of the datasets.
(2) The method proposed in this paper is more like a simple patchwork combination that lacks sound and rigorous theoretical support. Moreover, the paper lacks a more detailed and visual explanation of the proposed method.
(3) The article lacks experimental validation of the effectiveness of the various components of the method. For example, the effect of redundant background information in deep image encodings was not verified.
(4) The article also lacks a comprehensive analysis of the method's limitations, which would have facilitated a discussion of any potential challenges or failures. For example, the sub-optimal performance of the method proposed on the STC and Avenue datasets and the reasons for this should be analysed.

**Questions:**

1. How did you distinguish between normal and abnormal activity in multiple scenarios when constructing the two new datasets? What criteria were used?
2. How did you extract the human pose estimation, object velocity and depth image coding, can you provide more details on these?
3. How did you synthesise both few-frame and multi-frame features?
4. What are the limitations or failure cases of the MFAD method?
5. How did you verify the impact of each extracted feature on the results?
6. What are the limitations or failure cases of the MFAD method? Why the proposed method is sub-optimal for experiments on the STC and Avenue datasets?

---

> ### Author Response · Authors · 2023-11-20
>
> Thank you for your detailed review and insightful inquiries. We appreciate the opportunity to address your points of concern.
>
>
> Dataset Description:
>
> In Section 3 of our paper, we provide a comprehensive overview of our datasets—HMDB-AD and HMDB-Violence. These datasets are derived from the HMDB51 action recognition dataset and consist of short videos carefully trimmed around specific actions. HMDB-AD encompasses two normal behaviors, walking and running, widely used as fundamental normal human activities in VAD studies. The abnormal classes in HMDB-AD include climbing and performing a cartwheel, distinct activities characterized by varied orientation and movement direction. On the other hand, HMDB-Violence comprises nine normal non-violent activities and seven abnormal violent actions. In both datasets, we also include some randomly picked normal videos and add them to the test set.
>
> Appendix A showcases examples from both datasets to offer a clearer understanding.
>
>
> Methodology:
>
> Our methodology draws upon the principles established in AI-VAD. In their research, AI-VAD showcased the additional benefits of incorporating pose estimation, object velocity, and deep image encoding. To clarify our feature extraction pipeline, we offer a visualization in Figure 1 and an explanation in Section 4.1.
>
>
> Feature Synthesis:
>
> Following feature extraction, we compute anomaly scores for each feature independently and aggregate the results. This approach allows us to address the variations in dimensionality and scale across the four distinct features. We take note of this process in Section 4.2, supported by a visualization in Figure 2.
>
>
> Feature Extraction Details:
>
> We employ AlphaPose for pose estimation, derive object velocity through optical flows computed via FlowNet2, and utilize YOLOv3 for object detection.For deep image encoding, we leverage CLIP, using a ViT-32 backbone. We revised our paper to include these details in Section 5.2.
>
>
> Performance on ShanghaiTech and Avenue Datasets:
>
> MFAD sub-optimal performance on the ShanghaiTech and Avenue datasets is in line with our shift in approach to anomaly detection. These datasets primarily showcase a narrow definition of anomalies, whereas our methodology aims for a broader spectrum that also includes complex, multi-frame anomalies, as demonstrated in our proposed datasets. Consequently, MFAD faces similar challenges to previous methods when evaluated against these specific datasets and achieves comparable results. Particularly, the object-oriented aspect of MFAD struggles when confronted with scenarios involving closely clustered pedestrians. In contrast, as illustrated in Figure 3, our method outperforms previous methodologies when handling the limited subset of complex anomaly videos in the benchmark datasets.
>
> Following your note we addressed this shortly in Section 5.3.
>
>
> We trust that these clarifications shed light on your inquiries concerning our datasets, methodology, and the challenges faced with certain datasets. We remain open to further discussions and value your feedback immensely.
> Thank you for your engagement with our work and your insightful review.

---

### Official Review · Reviewer_y6mV · 2023-11-01

**Soundness:** 3 good
**Presentation:** 3 good
**Contribution:** 3 good
**Rating:** 6
**Confidence:** 4

**Summary:**

The manuscript underscores the importance of Video Anomaly Detection (VAD) in surveillance systems. It criticizes the current focus on simple, single-frame anomalies in benchmark datasets and advocates for expanding the scope of VAD to intricate anomalies. The authors introduce two datasets, HMDB-AD and HMDB-Violence, to challenge models with diverse action-based anomalies, and present Multi-Frame Anomaly Detection (MFAD). MFAD builds upon the AI-VAD framework, incorporating single-frame and two-frame features and applying density estimation. To tackle complex multi-frame anomalies, deep video encoding, and logistic regression are added. Experimental results highlight limitations in existing models with new anomaly types, demonstrating MFAD's proficiency in both simple and complex anomaly detection scenarios.

**Strengths:**

Strengths of the MFAD approach:

++ Comprehensive Feature Extraction: MFAD extracts four diverse feature types, including object velocities, human pose estimations, deep image encodings, and deep video encodings, enabling a holistic analysis of video data.

++ Adaptive Density Score Calculation: Using Gaussian Mixture Models (GMM) for velocity features and k-nearest neighbors (kNN) for other high-dimensional features, it adapts the density score calculation to the nature of the features, enhancing anomaly detection accuracy.

++ Max Feature Aggregation: The addition of the 'max' feature, which aggregates maximum feature scores per frame, adds value to the approach, improving anomaly detection.

++ Gaussian Smoothing: The application of Gaussian smoothing to anomaly scores reduces noise and provides more stable and interpretable results.

Overall, MFAD's strengths lie in its feature diversity, multi-modal analysis, adaptive density scoring, effective feature fusion, supervised learning, and robust experimental design, making it a powerful method for detecting both simple and complex anomalies in video data.

**Weaknesses:**

Looking at the manuscript, weaknesses are provided below.

-- Complexity: MFAD's multi-stage process and diverse feature extraction can make it computationally demanding and challenging to implement in resource-constrained environments.

-- Model Specificity: Utilizing specific video foundation models may reduce adaptability to different datasets or domains.

-- Not Real-Time: Computationally intensive and a requirement for separate training/testing data make real-time application challenging.

-- Gaussian Smoothing Limitation: Applying Gaussian smoothing may not suit all anomaly patterns, potentially leading to information loss.

**Questions:**

There are just a couple of questions that I need clarification on!!

--> How does MFAD handle the challenge of real-time video anomaly detection given its computational complexity? As I can see the author has provided the note for reproducibility, some insights would be helpful to understand the scope.
--> Can MFAD adapt to different video datasets and domains effectively, or is it limited by its reliance on specific video foundation models?

Currently, I'm leaning towards accepting this work, if the Authors can provide some insights into the weakness & questions section, that would be helpful to understand the significant contribution.

---

> ### Author Response · Authors · 2023-11-20
>
> Thank you for the comprehensive review and your thoughtful evaluation of our work. Your feedback is very important to us.
>
>
> Complexity:
>
> Our method's multi-stage process and diverse feature extraction are aligned with the state-of-the-art Video Anomaly Detection (VAD) methods [1]. The reliance on established models ensures that our method is comparable in run time to various prior approaches [2, 3].
>
> Foundation models:
>
> Foundation models are designed to be versatile and applicable across diverse domains [4]. Specifically, VideoMAEv2  [5] demonstrated their adaptability across various domains (Video Action Classification, Spatial Action Detection and Temporal Action Detection), and tested on multiple datasets.
>
>
> Real-Time Constraint:
>
> We acknowledge that real-time remains a challenge in current VAD research. As part of our future endeavors, we're exploring strategies to enhance computational efficiency. Our approach's modular structure enables parallel computation of individual features, offering a pathway to expedite processing. Furthermore, it provides our system an option for easily integrating future improved components that can speed up our method even more.
>
>
> We hope these insights address your inquiries and concerns. Thank you again for the thorough review and the time taken to consider our work.
>
>
> [1] Attribute-based Representations for Accurate and Interpretable Video Anomaly Detection
>
> [2] SSMTL++: Revisiting self-supervised multi-task learning for video anomaly detection
>
> [3] A Hybrid Video Anomaly Detection Framework via Memory-Augmented Flow Reconstruction and Flow-Guided Frame Prediction
>
> [4] On the Opportunities and Risks of Foundation Models. Rishi Bommasani et al
>
> [5] VideoMAE V2: Scaling Video Masked Autoencoders with Dual Masking

---

### Author Response · Authors · 2023-11-20

We're grateful to the reviewers for investing their time in reviewing our work and their efforts to improve it. We appreciate their acknowledgment of the method's effectiveness and noteworthy results (R1, R2, R3), recognition of the comprehensive experiments (R1, R3), and the positive feedback on the paper's clarity and structure (R2, R3, R4). We've addressed each of the concerns and questions individually and revised the paper accordingly.

---

### Meta-Review · Area_Chair_7eu4 · 2023-12-06

**Metareview:**

The paper aims to advance the existing video anomaly detection research to encompass intricate anomalies that extend beyond conventional benchmark boundaries. To this end, it introduces two datasets, HMDB-AD and HMDB-Violence, to challenge models with diverse action-based anomalies. A Multi-Frame Anomaly Detection (MFAD) model is developed that extends the existing AI-VAD framework  by adding a deep video encoding features capturing long-range temporal dependencies and logistic regression to enhance
final score calculation.

Some key concerns from the reviewers include (1) limited overall contribution as the proposed method appears to be a straightforward extension of an existing model, (2) missing of important and commonly used anomaly video datasets (e.g., UCF-Crime), and high time complexity that limits the potential application of the proposed method in real-time settings. Reviewers have also pointed out some limitations in the evaluation, including lack of a systematic ablation study to assess the effectiveness of the various components.

**Justification For Why Not Higher Score:**

The overall contribution is not adequately justified and evaluation does not appear to be convincing, either.

**Justification For Why Not Lower Score:**

N/A

---

### Decision · Program_Chairs · 2024-01-16

Reject